# Salinity Stress-Mediated Suppression of Expression of Salt Overly Sensitive Signaling Pathway Genes Suggests Negative Regulation by *AtbZIP62* Transcription Factor in *Arabidopsis thaliana*

**DOI:** 10.3390/ijms21051726

**Published:** 2020-03-03

**Authors:** Nkulu Kabange Rolly, Qari Muhammad Imran, In-Jung Lee, Byung-Wook Yun

**Affiliations:** 1Laboratory of Plant Functional Genomics, School of Applied Biosciences, Kyungpook National University, Daegu 702-701, Korea; rolly.kabange@gmail.com (N.K.R.); mimranbot@gmail.com (Q.M.I.); 2National Laboratory of Seed Testing, National Seed Service, SENASEM, Ministry of Agriculture, Kinshasa 904KIN1, Congo; 3Laboratory of Crop Physiology, School of Applied Biosciences, Kyungpook National University, Daegu 41566, Korea; ijlee@knu.ac.kr

**Keywords:** *AtbZIP62*, transcription regulation, SOS signaling pathway, salt tolerance, *Arabidopsis*

## Abstract

Salt stress is one of the most serious threats in plants, reducing crop yield and production. The salt overly sensitive (SOS) pathway in plants is a salt-responsive pathway that acts as a janitor of the cell to sweep out Na^+^ ions. Transcription factors (TFs) are key regulators of expression and/or repression of genes. The basic leucine zipper (bZIP) TF is a large family of TFs regulating various cellular processes in plants. In the current study, we investigated the role of the *Arabidopsis thaliana*
*bZIP62* TF in the regulation of SOS signaling pathway by measuring the transcript accumulation of its key genes such as *SOS1*, *2*, and *3*, in both wild-type (WT) and *atbzip62* knock-out (KO) mutants under salinity stress. We further observed the activation of enzymatic and non-enzymatic antioxidant systems in the wild-type, *atbzip62, atcat2* (lacking catalase activity), and *atnced3* (lacking 9-*cis*-epoxycarotenoid dioxygenase involved in the ABA pathway) KO mutants. Our findings revealed that *atbzip62* plants exhibited an enhanced salt-sensitive phenotypic response similar to *atnced3* and *atcat2* compared to WT, 10 days after 150 mM NaCl treatment. Interestingly, the transcriptional levels of *SOS1*, *SOS2*, and *SOS3* increased significantly over time in the *atbzip62* upon NaCl application, while they were downregulated in the wild type. We also measured chlorophyll a and b, pheophytin a and b, total pheophytin, and total carotenoids. We observed that the *atbzip62* exhibited an increase in chlorophyll and total carotenoid contents, as well as proline contents, while it exhibited a non-significant increase in catalase activity. Our results suggest that *AtbZIP62* negatively regulates the transcriptional events of *SOS* pathway genes, *AtbZIP18* and *AtbZIP69* while modulating the antioxidant response to salt tolerance in *Arabidopsis*.

## 1. Introduction

Salinity stress is one of the major challenges threatening food security and restricting crop productivity [1,2,3,4]. This abiotic stress affects the agricultural sector on a global scale. Salt stress induction leads to impaired plant growth and development (i.e., reduced rate of leaf surface expansion, cessation of expansion with the intensification of stress), crop failure, and cytotoxicity due to excessive ions, such as from sodium uptake and nutritional imbalance [5,6]. To cope with salt stress conditions, plants acquired diverse mechanisms that function to maintain a balanced reduction-oxidation state, reduced cellular hyperosmolarity, and ion disequilibrium [7]. The adverse effect of salinity stress is perceived at the whole-plant level. The signs are expressed throughout development, including germination, seedling stage, vegetative growth, and flowering. Salt stress involves the induction of signals that are influenced by diverse signaling pathways [8,9,10,11,12].

Leonard and Hepler [13] supported that the balance of calcium (Ca^2+^) and sodium (Na^+^) ions is an important component of salt tolerance in plants. Diverse physiological, molecular, and biochemical aspects of Na^+^ and Cl^−^ uptake and transport associated with salt stress were identified. Generally, during salinity stress, plants generate reactive oxygen species (ROS) [14] and reactive nitrogen species (RNS) [15], which can lead to oxidative damage [16] and lipid peroxidation [17], among other effects. To tackle the extent of oxidative or nitrosative stress, plants activate antioxidant systems that include accumulation of nitric oxide, induction of catalase, superoxide dismutase, and peroxidase activity, enhancement of chlorophyll and proline content, etc. [17,18,19,20,21,22]. In order to withstand disturbed environmental conditions, such as salt stress, plants activate genes and transcription factors (TFs) coding genes. TFs are proteins indispensable for regulating gene expression and, as such, TFs modulate essential aspects of organismal function. Intrinsic to the TF mode of action is their ability to interact with particular sequences of DNA, as well as other proteins as part of their transcriptional activities to regulate gene expression [23]. ROS or RNS generation, transcriptional regulation, and ionic homeostasis are interconnected; therefore, they are part of the complex network modulating the response of plants, including salinity tolerance, to abiotic stress [24,25].

The salt tolerance mechanisms involve, in addition to the various antioxidant systems, the activation of many other biosynthetic and signaling pathways such as the salt overly sensitive (SOS) pathway [12]. In plants, the SOS pathway comprising *SOS1*, *SOS2*, and *SOS3* was suggested to regulate cellular signaling during salt stress to mediate ion homeostasis. It is well established that SOS pathway genes are involved in Na^+^ extrusion [26].

In rice [27,28] and *Arabidopsis* [29,30], genes encoding the SOS proteins were identified. These genes play an important role in maintaining a balanced ion level within the cell and in conferring salt tolerance. Moreover, an increase in salt tolerance was recorded in plants overexpressing SOS genes [31]. The *Arabidopsis SOS* genes (*1* to *3*) co-express and are involved in the adaptive response to salt tolerance [29,31].

The SOS pathway in rice was shown to correlate with the ability to exclude sodium from the shoot and maintain a controlled and low Na^+^/K^+^ ratio [32,33]. Upon salt stress induction, SOS pathway genes are reported to be induced, and they regulate vacuolar Na^+^/H^+^ exchange in *Arabidopsis* [34]. These genes play an important role in pumping sodium ions out of the cell, among others, in addition to interacting with calcium, mitogen-activated protein kinases (MAPKs), and calcium-dependent protein kinases, as well as being involved signaling and signal transduction. MAPKs are protein serine/threonine kinases, which convert the extracellular stimuli into broad-spectrum cellular responses [35]. In addition, the characterization of tonoplast (vacuolar membrane) Na^+^/H^+^ exchange suggested that the activity of the regulatory pathway components (SOS1) originated from the AtNHX proteins [36]. Moreover, TFs such as bZIP were reported as being part of the adaptive response mechanisms to salt tolerance in plants [37,38,39,40].

In order to investigate the role of the *AtbZIP62* gene encoding TF in the regulation of expression of SOS signaling pathway genes under salinity stress in plants, this study monitored the transcriptional response of three well-identified *Arabidopsis thaliana* SOS pathways genes in both wild-type (WT) and the *atbzip62* mutant exposed to NaCl-induced salt stress. Additionally, changes in physiological and biochemical processes caused by the perturbation in the *AtbZIP62* under salt stress, activation of antioxidants systems, modification of chloroplast pigment composition, proline content, lipid peroxidation, and the phenotypic response of *atbzip62* under salinity stress were studied to elucidate the function of *AtbZIP62* TF in salinity stress response. Moreover, the *atcat2* deficient in *AtCATALASE2* [41] and *atnced3* deficient in *AtNCED3* (9-*cis*-epoxycarotenoid dioxygenase involved in ABA pathway) [42] KO mutants were included in this study as salt-sensitive mutants to study the role that *AtbZIP62* could play in maintaining a balanced cellular reduction–oxidation state within the cell during salinity stress.

## 2. Results

### 2.1. Initial Screening for Selection of NaCl Concentration

Prior to exposing *Arabidopsis* plants to salt stress at the rosette stage, we performed an initial screening of four genotypes to evaluate their response to salinity stress on MS media modified with gradient NaCl concentrations of 50, 100, 150, or 200 m*M*. We evaluated their germination ability for five days. The results showed that, under 150 m*M* NaCl, the germination of *atbzip62* seeds was inhibited significantly compared to WT, *atnced3*, and *atcat2* (see Appendix A). Other NaCl concentrations (50 m*M* and 100 m*M*) also showed various reductions in germination patterns, after which recovery was observed. However, 200 m*M* completely inhibited the germination of *atbzip62* seeds. To select the NaCl concentration for downstream experiments, we cross-evaluated the germination frequency data, as well as the shoot and root growth patterns. The cotyledon development frequency (CDF) was calculated by counting only the developed green seedlings (as some seeds might germinate but do not survive on stress media after long exposure) as described previously [43,44] of the targeted genotypes on media with different concentrations of salt. We identified 150 m*M* NaCl as the most effective concentration for downstream experiments, due to the inhibitory effect recorded (Figure 1).

### 2.2. Growth of Arabidopsis Genotypes on Gradient NaCl Concentrations

We were interested to see how the *atbzip62* KO mutant would respond to salt stress (Figure 1). The results showed that low salt concentration (50 m*M*) did not affect the growth of *Arabidopsis* genotypes two weeks after exposure. However, when grown on 100 m*M* NaCl, *atbzip62*, *atnced3*, and *atcat2* were affected significantly compared to WT. Furthermore, when grown on 150 m*M*, the survival rate of all genotypes dropped drastically. When 200 m*M* was supplied, none of the genotypes grew. We further observed that shoot growth was inhibited significantly in response to gradient NaCl concentrations (Appendix A). The maximum inhibitory effect of NaCl-induced salt stress was recorded under 200 m*M* on MS medium, where total shoot growth inhibition was observed in the Col-0 wild type and *atbzip62* mutant. In contrast, under 150 m*M* treatment, all genotypes had their shoot growth inhibited, but not completely. Low NaCl concentration (50 m*M*) did not have an inhibitory effect on root growth; rather, we observed a slight increase in root length (Appendix A). From 100 m*M*, plants started experiencing a reduction in root growth. The shortest root length was recorded in *atbzip62*. Furthermore, when grown on 150 m*M* NaCl, all genotypes exhibited strong inhibition of root growth. However, 200 m*M* NaCl inhibited root growth completely.

### 2.3. The Arabidopsis Atbzip62 Loss-of-Function Mutant Is Sensitive to Salt Stress

Upon exposure to high salinity, four *Arabidopsis* genotypes exhibited different levels of response (Figure 2). For instance, Col-0 showed some level of tolerance. The leaves were intact at 10 days after salt stress induction. In contrast, *atbzip62* showed severe leaf drying, wilting, and high sensitivity; *atnced3*, lacking the *AtNCED3* gene, and *atcat2*, lacking the *AtCAT2* gene, were affected severely by 150 m*M* NaCl.

### 2.4. Arabidopsis AtbZIP62 Loss-of-Function Mutant Upregulated AtPYD1, AtbZIP18, and AtbZIP69 under Salt Stress

After exposure of the *atbzip62* loss-of-function mutant to high salinity, we investigated the transcriptional levels of *AtPYD1* by quantitative real-time PCR (qRT-PCR), the ortholog of the rice *DHODH1* observed previously to be upregulated by drought stress and salinity [45], in the *atbzip62* mutant background. The results revealed that *AtPYD1* expression was upregulated significantly in *atbzip62*, and increased over time, while it was downregulated in WT under the same conditions (Figure 3A). Under NaCl stress conditions, *AtPYD1* showed a contrasting expression pattern (upregulated in WT and downregulated in *atbzip62*). A slight increase was also observed when *AtPYD1* was expressed in *atcat2*, lacking the *AtCAT2* gene. We were also interested to see the pattern of *AtbZIP62* expression in WT under conditions of varying salinity. The qPCR results revealed that the expression of *AtbZIP62* was upregulated significantly in *atcat2* compared to WT (Figure 3B). We also observed that *AtbZIP18* (Figure 3C) and *AtbZIP69* (Figure 3D) were upregulated similarly in the *atbzip62* loss-of-function mutant under the same conditions.

### 2.5. The Expression of SOS Signaling Pathway Genes Is Upregulated in atbzip62 Loss-of-Function Mutant Exposed to Salt Stress

SOS pathway genes regulate the balance of salt ions in the cell during salinity stress. This evidence helped us investigate the role of *AtbZIP62* TF in the adaptive mechanism of salt tolerance in plants. We expressed three SOS pathway genes in *atbzip62* exposed to NaCl-induced salt stress. Interestingly, all three SOS genes were downregulated similarly in the WT, whereas, in *atbzip62*, they were upregulated significantly (Figure 3E–G).

### 2.6. Glutamate Synthase-Encoded Genes Are Induced by Salt Stress in atbzip62

Glutamate synthase is involved in the initial steps of nitrogen assimilation in plants. Abiotic stresses do not only prevent plants from obtaining available water, but they also restrict them from acquiring essential nutrients, such as nitrogen, for their growth and development. Therefore, we investigated the transcriptional regulation of high-affinity nitrogen assimilation genes related to *AtbZIP62* under salt stress. Our data, presented in Figure 3H, show that the glutamate synthase-encoding gene 1 (*Gls1*) was downregulated by salt stress in Col-0, whereas it was upregulated significantly in the *atbzip62* KO mutant. Similarly, expression of the gene encoding glutamate synthase 2 was suppressed in Col-0 (Figure 3I).

### 2.7. Change in Chloroplast Pigment Content under Salt Stress

Chlorophylls are major pigments that accumulate in the chloroplast and are key players in light-harvesting during the photosynthetic process. Chlorophyll *a* (Figure 4A), chlorophyll *b* (Figure 4B), and chlorophyll *a* + *b* (Figure 4C) were shown to decrease significantly in Col-0 wild type after exposure to NaCl-induced salinity over time. However, in *atbzip62*, a significant increase was recorded at 3 h and a decrease was recorded at 6 h. In *atnced3*, all chlorophylls decreased over time in response to salt stress in the same manner as observed in WT. Total carotenoid content exhibited patterns similar to chlorophylls under salt stress (Figure 4D).

Pheophytin is important in the transfer of electrons during photosynthesis. Here, we observed that pheophytin *a* pigment decreased significantly in WT 6 h after salt stress. In the *atbzip62* and *atnced3* loss-of-function mutants, we recorded an opposite pattern, where a significant increase in pheophytin *a* was observed after 3 h (Figure 4E). Pheophytin *b* decreased over time in WT, *atbzip62,* and *atnced3* (Figure 4F). The level of pheophytin *a* + *b* (Figure 4G) and total carotenoids relative to pheophytin *a* and *b*, referred to as C*x + c* (Figure 4H), in response to salinity stress decreased significantly over time in WT and *atnced3*, whereas a significant increase in the levels of each was noted 3 h after stress induction in *atbzip62*.

### 2.8. Detected Antioxidant Enzyme Activity, Proline Accumulation, Change in Protein, and Malondialdehyde (MDA) Content under Salt Stress in Arabidopsis Genotypes

In order to investigate the physiological processes activated during the exposure of *Arabidopsis* genotypes to salt stress, we measured the activity of antioxidant enzymes and the accumulation of proline. A gradual and persistent increase in catalase (CAT) activity was detected in WT plants from 0 h to 3 h to 6 h of exposure (Figure 5A). CAT activity also increased in *atbzip62* at 3 h, but decreased slightly by 6 h. Both *atbzip62* and *atcat2* exhibited a similar pattern of CAT activity over time. Peroxidase (POD) activity (Figure 5B) decreased over time in WT. A slight increase in POD activity was observed in *atbzip62* after 6 h of exposure to salt stress. A significant increase in POD activity was recorded in *atnced3* and *atcat2* after 3 h of exposure to salt stress, and a decrease was observed at 6 h in *atcat2*. As for polyphenol oxidase (PPO) activity (Figure 5C), WT and *atbzip62* exhibited a decrease at 3 h. At 6 h, PPO activity slightly increase in WT, while slightly decreasing in *atbzip62.* PPO activity for *atnced3* increased over time and *atcat2* experienced an exponential increase in PPO from 0 h to 3 h to 6 h after salt stress.

The primary response of plants to abiotic stimuli is a decrease in regular metabolic activities. Consequently, there is a reduction in growth. When a reduction in growth occurs, the synthesis of protein is the most negatively affected anabolic process [46]. Figure 5D shows a significant increase in protein in *atbzip62* at 3 and 6 h compared to WT, which shows a similar pattern from 0 h to 3 h to 6 h after NaCl application. A steady increase in protein content was seen in *atnced3* over time, whereas *atcat2* recorded an increase in proteins at 3 h, but a decrease after 6 h.

Proline, unlike other amino acids, was shown to accumulate in a particular manner in response to abiotic stress, such as drought and salinity. Here, we report that proline content significantly increased in *atbzip62* 3 h after NaCl application compared to WT and then decreased at 6 h (Figure 5E). In contrast, the level of proline accumulation recorded in *atnced3* was significantly lower than that in WT and *atbzip62* under the same conditions. However, the *atcat2* exposed to salt stress by NaCl recorded a similar proline accumulation pattern to WT at 3 h of exposure and remained at the same level at 6 h.

We further investigated lipid peroxidation and cell membrane degradation by measuring malondialdehyde (MDA) accumulation in plants exposed to salt stress at the rosette stage. The results (Figure 5F) indicate that, upon the exposure of WT, *atbzip62*, and *atnced3* loss-of-function mutants to salt stress induced by 150 m*M* NaCl, the level of MDA increased significantly in WT compared to non-stressed plants. In contrast, *atbzip62* plants exhibited a slight decrease in MDA content and significant decreased in *atnced3*.

## 3. Discussion

### 3.1. AtbZIP62 Regulates SOS Pathway Genes under Salt Stress Conditions

When plants are grown in a saline environment, a high accumulation of Na^+^ occurs in the cell cytoplasm, which leads to disruption of metabolic processes and growth reduction. The coordination of transport proteins on cellular membranes is of great importance to maintaining a balanced level of cytoplasmic sodium. Some evidence supports the idea that SOS pathway genes *SOS1*, *SOS2*, and *SOS3* are linked to salt tolerance in *Arabidopsis* [47]. The activity of SOS1, the plasma membrane Na^+^/H^+^ exchanger, was shown to be regulated by SOS2, a protein kinase [48], and SOS3, a calcium-binding protein [34].

SOS pathways genes were proposed to mediate signaling in response to salinity in plants while maintaining ion homeostasis [49,50]. In addition, the SOS regulatory pathway was shown to be the target for Na^+^/H^+^ exchange in the tonoplast (vacuolar membrane) of *Arabidopsis*, which was supported by fluctuations in Na^+^/H^+^ exchange activity in the tonoplast of the mutants (*sos1, sos2, sos3*) compared to the wild type. In the present study, all three *Arabidopsis SOS* genes were significantly upregulated over time in the *atbzip62* loss-of-function mutant and downregulated in the WT. The expression level of *SOS1* was significantly higher than that of *SOS2* and *SOS3*. A recent study of the plasma membrane Na+/H+ exchanger reported that the rice *SOS1* gene plays a pivotal role in the adaptive response to salt tolerance [51]. It is thought that *Arabidopsis SOS1* could play a leading role in modulating ion pumping and exchange within the cell during salt stress events. However, despite being involved in the positive regulation of plant responses to salinity by controlling the Na^+^/H^+^ exchange rate, *SOS1, SOS2,* and *SOS3* transcript accumulations were found to be negatively regulated by *AtbZIP62* TF per the transcriptional patterns recorded in the *atbzip62* loss-of-function mutant exposed to salt stress conditions. This is consistent with a previous report [49], supporting the idea that many genes control the salt tolerance trait in plants as part of a complex regulatory network. Another study [52], focusing on the transcriptional response of the *SOS1* gene in *Aeluropus littoralis,* found that the messenger RNA (mRNA) level of *SOS1* increased in response to 200 and 400 m*M* NaCl. In a similar study by Martínez-Atienza, et al. [53], mutant plants deficient in *SOS1, SOS2,* or *SOS3* shared similar salt-sensitive phenotypes, and their co-expression was shown to improve salt tolerance [31]. Our study suggests that *AtbZIP62* could be required for the transcriptional regulation of *Arabidopsis SOS* signaling pathway genes while modulating the adaptive response to salt tolerance, which includes the regulation of ion exchange, signal perception and transduction, and modulation of assimilation.

Hence, using the expression data, this study proposes a simplified model in Figure 6 illustrating the possible interactions between the *AtbZIP62* TF and the SOS pathway genes, which would involve two other *AtbZIP* TF family members, *AtbZIP18* and *AtbZIP69*, as part of the plant adaptive response to salt tolerance.

### 3.2. AtbZIP62 Is Involved in the Regulation of Antioxidants Systems and Chloroplast Pigment Accumulation under Salt Stress

Generally, the adaptive response of plants to abiotic stress such as salinity involves the activation of antioxidant systems. This response involves the perception and transduction of signals resulting in the accumulation of enzymes such as CAT, and stress amino acids like proline, among others. Therefore, the recorded CAT activity in all genotypes exposed to high salinity, the differential activity in POD and PPO, and the proline accumulation in the WT compared to *atbzip62* loss-of-function mutant indicate that the transcription factor-coding gene *AtbZIP62* could play an important role in the regulation of antioxidant systems under salinity stress.

Abiotic stress induces ROS accumulation, leading to oxidative stress, which results in oxidative damage to plants. The persistence of the oxidative stress may cause cell membrane degradation and lipid peroxidation. Our data showed that the WT plants exhibited a significant increase in MDA content under salinity stress. Unexpectedly, MDA accumulation in *atbzip62* loss-of-function mutant slightly decreased upon salt stress induction. In general cases, the level of lipid peroxidation is expected to be high over time in sensitive genotypes. In this study, *atbzip62* was shown to be a highly sensitive phenotype in response to salt stress 10 days after exposure (Figure 2). We mentioned previously that the expression of *AtbZIP62* was downregulated over time in WT exposed to NaCl-induced salt stress. These findings suggest that *AtbZIP62* contributes to the maintenance of a balanced cellular redox state, maintaining cell membrane integrity and controlling the level of lipid peroxidation under salinity stress.

Carotenoids are the second most abundant group of pigments in plants [54], found in photosynthetic and non-photosynthetic tissues. Their yellow or orange pigments and, on occasion, red color characterize the carotenoids found in photosynthesizing cells. The data presented in Figure 4 show that, regardless of the difference in the accumulation levels in the tested *Arabidopsis* genotypes, the pattern of carotenoid content was similar to the pattern of chlorophylls at all time points, thus supporting the suggestion that the color of carotenoids in the cells is masked by chlorophyll, which gives plants their characteristic green color. Furthermore, *AtbZIP62* is proposed to play a role in the regulation of chloroplast pigments under salinity stress in *Arabidopsis*.

## 4. Materials and Methods

### 4.1. Plant Materials and Screening for Salt Tolerance of atbzip62 on Gradient NaCl Concentrations

To perform the experiments, we used *Arabidopsis* Col-0 (wild type), *atbzip62*, *atnced3* (abscisic acid (ABA)-deficient mutant for the possible role of *AtbZIP62* in the ABA signaling-mediated salt stress response), and *atcat2* (high H_2_O_2_-producing mutant lacking the *AtCAT2* gene). Seed sterilization followed the protocol described previously [55].

We initially conducted a screening of the *atbzip62* loss-of-function mutant exposed to gradient NaCl concentrations on MS medium (50, 100, 150, or 200 m*M*) in order to identify the adequate concentration for downstream experiments. We tested seed germination capacity on salt-modified media for five days. For each genotype, 30 seeds were sown, 10 per replication. Seedlings were kept on salt medium to assess the growth ability under salinity stress conditions. The ability of *Arabidopsis* genotypes to survive on NaCl-stressed medium was evaluated 14 days after sowing. The cotyledon development frequency (CDF) was calculated by counting only the developed green seedlings (as some seeds might germinate but do not survive on stress media after long exposure) as described previously Seedlings with green, expanded, and intact cotyledons were counted, and CDF was calculated relative to the total number sown/germinated as described previously [44] and expressed as percentage cotyledon development frequency. On the 10th day of sowing, shoot and root length were measured.

### 4.2. Arabidopsis Genotypes Growth Conditions and Salt Stress Induction

Well-sterilized seeds were grown under growth chamber conditions (approximately 22 °C and a 16-h/8-h light/dark cycle). *Arabidopsis* seeds were sown on the surface of a peat moss–soil mixture with perlite and vermiculite in 50-well trays. A preventive antifungal treatment was given, mixed with the peat moss–soil to avoid fungal growth during the experimental period. Water was given via sub-irrigation daily during the first few days after sowing and reduced to twice a week after plants developed their two true leaves to provide adequate moisture for plant maintenance and growth until flowering.

To induce salt stress conditions, at least 30 *Arabidopsis* plants, 10 per replication, 150 m*M* NaCl, identified as the appropriate concentration after the screening, was applied at the rosette stage via sub-irrigation. Leaf samples were collected at 0 (plants supplemented with water only), 3, and 6 h (NaCl-treated plants) for gene expression analysis, enzyme activity assays, and chlorophyll, pheophytin, and proline measurement. However, for phenotypic observations, plants were exposed to NaCl-induced salt stress for 10 days, and the response of each genotype was evaluated.

### 4.3. Total RNA Isolation, Complementary DNA (cDNA) Synthesis, RT-PCR, and Real-time qPCR Analysis

Total RNA was isolated from leaf samples using the TRI-Solution^TM^ Reagent (Cat. No: TS200-001, Virginia Tech Bio-Technology, Lot: 337871401001) as described by the manufacturer. Thereafter, the complementary DNA (cDNA) was synthesized as described previously [56]. Briefly, 1 µg of RNA was used to synthesize cDNA using BioFACT^TM^ RT-Kit (BioFACT^TM^, Republic of Korea) according to the manufacturer’s standard protocol. The cDNA was then used as a template in reverse transcription PCR (RT-PCR) or real-time (RT) qPCR (RT qPCR) to study the transcript accumulation of selected genes (Appendix A). For RT-PCR, a reaction mixture of 7 µL of 2× F-Star Taq PCR Master mix (BioFACT, Korea) and 10 n*M* of each RT-PCR forward and reverse primer in a final volume of 25 µL was used. A three-step cycling reaction was performed, including polymerase activation at 95 °C for 2 min, 95 °C strand separation for 20 s, annealing at 56–58 °C for 40 s for 25 cycles, extension at 72 °C for 1 min/kb, and the final extension at 72 °C for 5 min.

For quantitative real-time reverse transcription PCR (qRT-RT-PCR), we prepared a reaction mixture compost of SYBR green (BioFACT, Korea) along with 100 ng of template DNA and 10 n*M* of each forward and reverse primer in a final volume of 20 µL. A no-template control was used. A two-step reaction including polymerase activation at 95 °C for 15 min, followed by denaturation at 95 °C for 5 s and annealing and extension at 65 °C for 30 s, was performed in a real-time PCR machine (Eco™ Illumina). Total reaction cycles were 40 and the data were normalized with relative expression of *Arabidopsis Actin*.

### 4.4. Proline Measurement Assay

Quantification of proline was done using the colorimetric method described previously [57]. Approximately 100 mg of leaves collected from well-watered or NaCl-treated plants of Col-0, *atbzip62*, *atnced3,* or *atcat2* were homogenized in 3% sulfosalicylic acid (5 µL mg^−1^ FW) in Eppendorf tubes. The homogenate was centrifuged at room temperature (benchtop centrifuge) at 13,000 rpm for 5 min. Then, 100 µL from the supernatant of plant extract was added to the reaction mixture (100 µL of 3% sulfosalicylic acid, 200 µL glacial acetic acid, 200 µL acidic ninhydrin (1.25 g ninhydrin (1,2,3-indantrione monohydrate)), 30 mL glacial acetic acid, 20 mL of 6 *M* orthophosphoric acid (H_3_PO_4_), dissolved into double distilled water and stored at 4 °C). The mixture was incubated at 96 °C for 1 h and cooled immediately on ice to terminate the reaction. The samples were extracted by adding 1 mL of toluene to the reaction mixture, vortexed for about 20 s, and allowed to rest for 5 min on the bench in order to permit separation of the organic and water phases. Then the chromophore (upper phase colored light red) containing toluene was moved into a fresh cuvette, and the absorbance was read at 520 nm using toluene as a reference. The proline concentration was calculated on the fresh weight basis and expressed in µg g^−1^ FW.

### 4.5. CAT, PPO, and POD Activity Assay

The activity of antioxidant enzymes was assayed using the spectrophotometric method as described previously [58]. Briefly, 100 mg of leaves were ground to a fine powder using liquid nitrogen and immediately homogenized in 50 m*M* phosphate buffer (pH 7.5). The mixture was centrifuged for 10 min at 12,000 rpm at 4 °C after being kept on ice for approximately 10 min. The supernatant was then transferred to fresh 1.5-mL e-tubes. The homogenate was used as a crude enzyme source for CAT, PPO, and POD activities.

CAT activity was assayed as described previously [59]. Briefly, a volume of 50 µL of H_2_O_2_ (50 m*M*) was added to the crude enzyme extract, and the absorbance of the reaction mixture was measured at 240 nm after 1 min. CAT activity was expressed per milligram of the sample’s fresh weight [60].

The spectrophotometric method was also used to estimate PPO and POD activities as described previously [61]. The supernatant obtained after centrifugation was used as a crude enzyme source. For POD, the reaction contained 50 µL of crude extract, 50 µL of pyrogallol, 25 µL of H_2_O_2_, and 10 µL of phosphate buffer, and it was incubated for 5 min in dark conditions. Then, 25 µL of H_2_SO_4_ was added to the reaction mixture. The absorbance was measured at 420 nm [62]. Calculations were done using the method described previously [63].

### 4.6. Total Protein Assay

The spectrophotometric Bradford assay [64] was used to estimate the total protein in *Arabidopsis* leaf extracts exposed to salinity. This method is based on the absorbance shift observed in an acidic solution of dye Coomassie^®^ Brilliant Blue G-250. When added to a solution of protein, the dye binds to the protein resulting in a color change from reddish-brown to blue. The absorbance of the complex protein-dye was measured at 595 nm (vs. the blank) about 5 min after initiation of the reaction [65]. The protein concentration was expressed as µg g^−1^ FW.

### 4.7. Lipid Peroxidation Assay

The lipid peroxidation level in the leaf tissue was measured as the MDA content, determined by the thiobarbituric acid (TBA) reaction [66]. About 200 mg of nontreated and treated samples were homogenized in 4 mL of trichloroacetic acid (TCA) (0.1%) with a porcelain mortar and pestle, followed by centrifugation for 15 min at 10,000 rpm. Then, 1 mL of the supernatant was harvested and 2 mL of TCA (20%) containing 0.5% TBA was added. The supernatant was incubated in a preheated water bath at 95 °C for 30 min and cooled immediately on ice. The reaction mixture was centrifuged for 10 min at 10,000 rpm, and the optical density (OD) was measured at 532 nm and 600 nm (OD_600_ is the nonspecific absorbance that is subtracted from the OD_532_ reading). MDA level was calculated using Lambert’s equation (extinction coefficient of MDA, 155 n*M*^−1^ cm^−1^).

### 4.8. Chlorophyll, Pheophytin, and Total Carotenoids Content Measurements

Chlorophylls, pheophytin, and total carotenoids were measured as described previously [67]. Approximately 200 mg of leaf tissue was homogenized with acetone (80%), followed by centrifugation for 10 min at 3000 rpm. The supernatant was transferred to fresh falcon tubes, and centrifugation was repeated until all chlorophyll was harvested in the solvent. The combined supernatant was made to known volume with acetone (80%). The absorbance of the extract was read at 645, 663, and 652 nm for chlorophyll *a*, *b*, and total chlorophyll, 480 and 510 nm for total carotenoids, and 665, 653, and 470 nm for pheophytin a, b, and total pheophytins, using acetone as a blank. The OD values at 645, 663, and 652 were used to calculate chlorophyll contents and total carotenoids as described previously [68].

### 4.9. Statistical Analysis

All experiments were done using a completely randomized design. Experiments were repeated two times. Data were collected in triplicate and analyzed statistically using GraphPad Prism software (Version 7.00, 1992–2016 GraphPad). Analysis of variance for completely randomized design was performed, and the least significant difference was calculated where needed at a significance level of 0.05. Where applicable, at any significant effect, differences between means were evaluated for significance by using the Student’s *t*-test.

## 5. Conclusions

The current study investigated the role the *Arabidopsis AtbZIP62* encoding a transcription factor in modulating the transcriptional events of the salt overly sensitive pathway genes in response to salt stress. The SOS pathway consist of three important genes *SOS1*, *SOS2*, and *SOS3*. Our results suggested that all three SOS genes, *SOS1*, *2*, and *3*, were significantly downregulated by NaCl-induced salt stress in the *atbzip62* background, with *SOS1* exhibiting the highest expression level. In addition, the transcriptional levels of *AtbZIP18* and *AtbZIP69*, two candidate bZIP TF encoding genes expressed in the *atbzip62* background, were significantly induced compared to wild type under the same conditions. Furthermore, the observed phenotypic response of *atbzip62* revealed an enhanced sensitivity toward salt stress, followed by an increase in the chlorophyll, pheophytin, and carotenoids contents, the activation of catalase activity, and the accumulation of malondialdehyde and proline. Therefore, this study suggests that *AtbZIP62* negatively regulates the transcript accumulation of SOS signaling pathway genes, *SOS1*, *SOS2,* and *SOS3*, as well as *AtbZIP18* and *AtbZIP69* in *Arabidopsis thaliana*, while maintaining a balanced redox state under salt stress conditions.

## Figures and Tables

**Figure 1 ijms-21-01726-f001:**
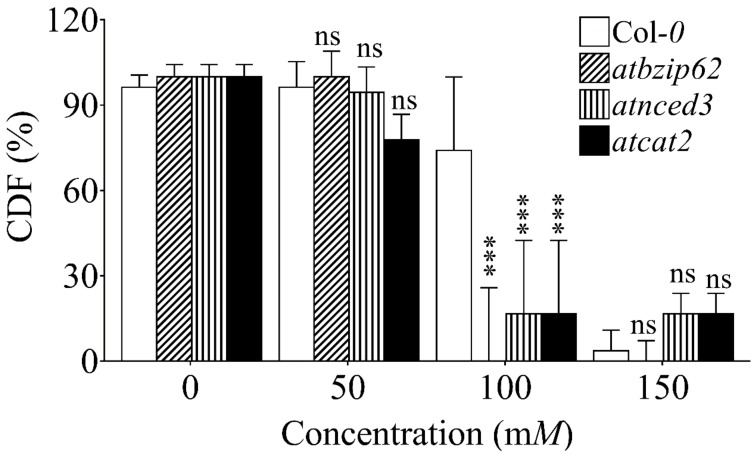
Growth of *Arabidopsis* mutant lines on gradient NaCl concentrations. Four *Arabidopsis* genotypes, the Col-0 wild type, the *atbzip62*, *atnced3*, and *atcat2* loss-of-function mutants were exposed to salt stress induced by gradient NaCl concentrations. The survival of all genotypes was recorded as a percentage of seedlings having green cotyledons over the total number of seeds germinated (referred to as cotyledon development frequency or CDF) on half-strength MS medium for two weeks. Data points are the mean values of triplicate assays while bars indicate ±SD. The statistical significance level is indicated by asterisks on top of each bar. *** *p* < 0.05 compared to the wild type; ns, non-significant.

**Figure 2 ijms-21-01726-f002:**
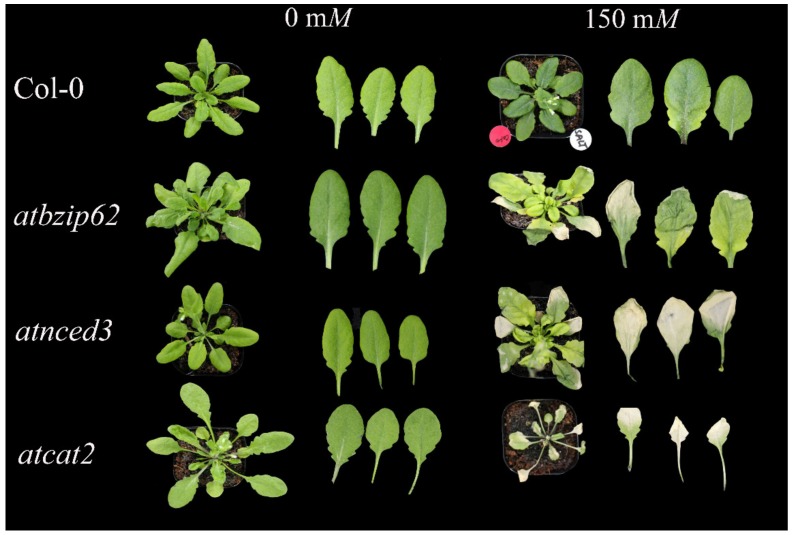
Phenotypic response of *Arabidopsis* genotypes to salt stress at the rosette stage 10 days after NaCl treatment. The left panel shows plants and detached leaves grown under normal conditions (no salt treatment), whereas the right panel shows plants subjected to salt stress conditions (150 m*M*). Pictures were taken 10 days after salt stress induction. Adobe Photoshop CS6 was used to remove the background for better display and visualization.

**Figure 3 ijms-21-01726-f003:**
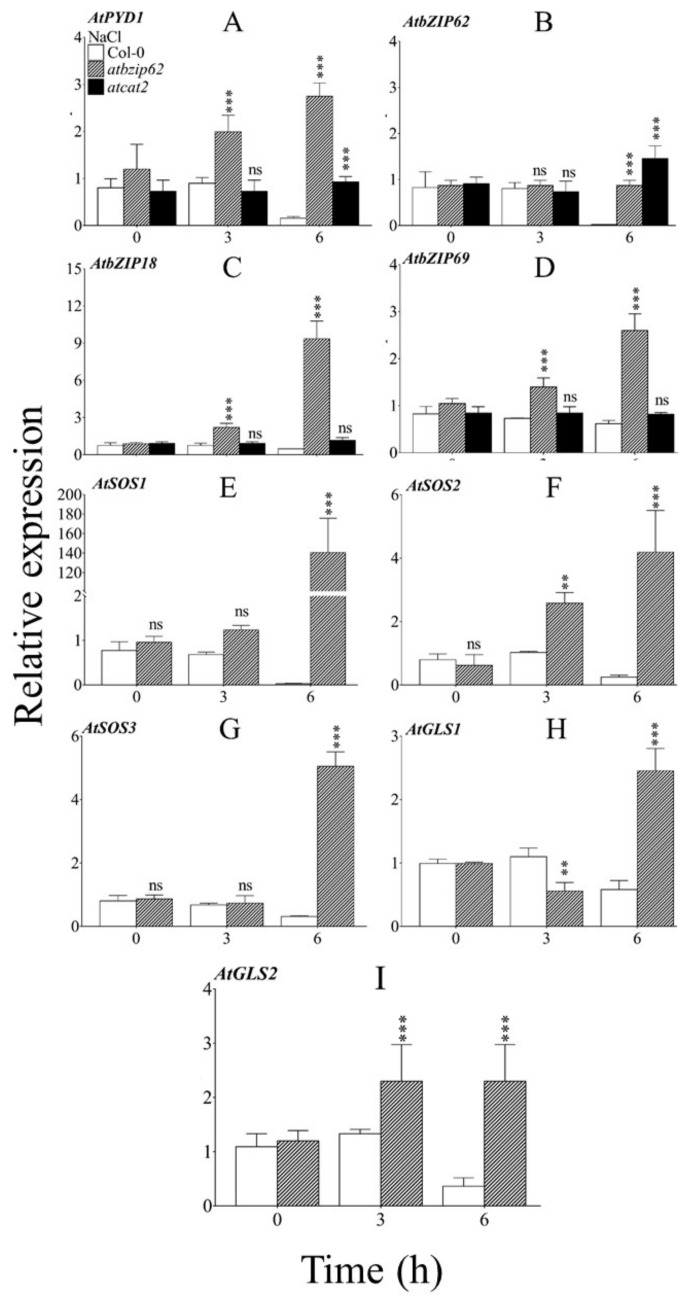
Transcripts accumulation of AtbZIP transcription factors (TFs) encoding genes, *AtPYD1*, SOS, and glutamate synthase under salt stress. (**A**) *AtPYD1* expression in response to salt stress (150 mM) for 3 to 6 h in Col-0 wild type, *atbzip62* loss-of-function mutant, and *atcat2*. (**B**) *AtbZIP62* transcript accumulation in Col-0, *atbzip62* loss-of-function mutant, and *atcat2*. (**C**) Expression pattern of *AtbZIP18* over time. (**D**) Expression pattern of *AtbZIP69* over time. (**E**) Expression of *SOS1* in Col-0 wild type and *atbzip62* loss-of-function mutant over time. (**F**) Transcriptional level of *SOS2*, (**G**) *SOS3*, (**H**) *AtGLS1*, and (**I**) *AtGLS2* overtime at the rosette stage in Col-0 and *atbzip62*. All the data points are the mean of triplicate assays, while error bars represent ± SD. Statistical significance is indicated by asterisks on top of bars. ** *p* < 0.01; *** *p* < 0.001 compared to wild type; ns, non-significant.

**Figure 4 ijms-21-01726-f004:**
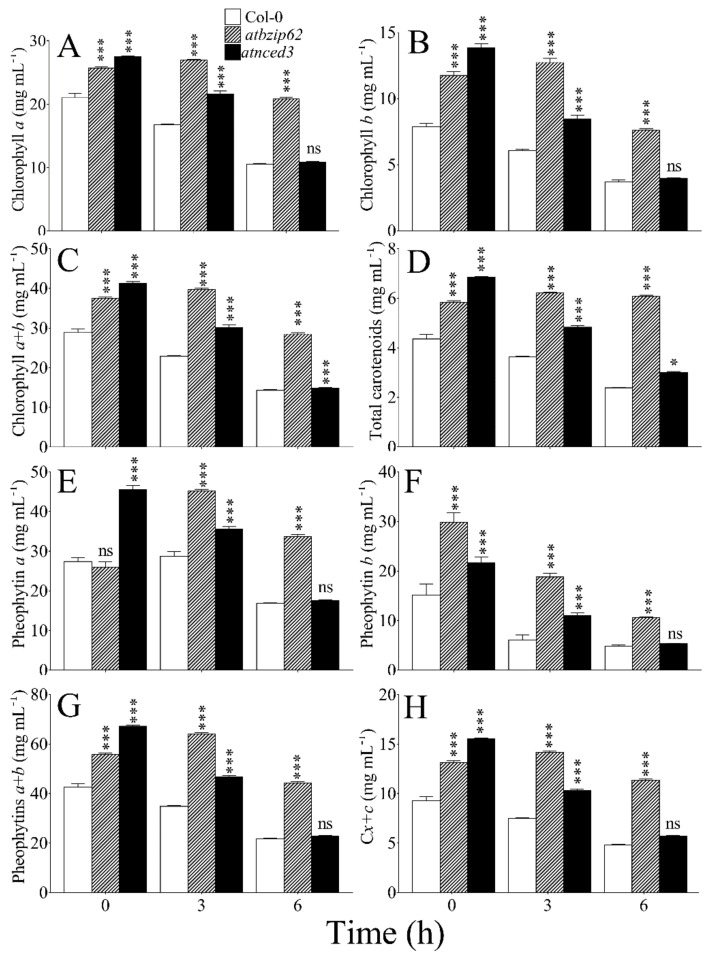
Pattern of chloroplast pigment content under salt stress. (**A**) Chlorophyll a content. (**B**) Chlorophyll b content. (**C**) Chlorophyll a + b content. (**D**) Total carotenoids. (**E**) Pheophytin a content. (**F**) Pheophytin b content. (**G**) Pheophytin a + b content. (**H**) Total carotenoids relative to pheophytin a and b (Cx + c). The first set of three bars in each panel represents time 0 h (no salt stress), the second set of three bars represents enzyme activity at 3 h post-stress induction, and the third set of three bars represents enzyme activity at 6 h post-stress induction. All the data points are the mean of triplicate assays, while error bar represents ± SD. The asterisks on top of bars are statistical significance level with 95% confidence level. * *p* < 0.05; *** *p* < 0.001 compared to wild type; ns, non-significant.

**Figure 5 ijms-21-01726-f005:**
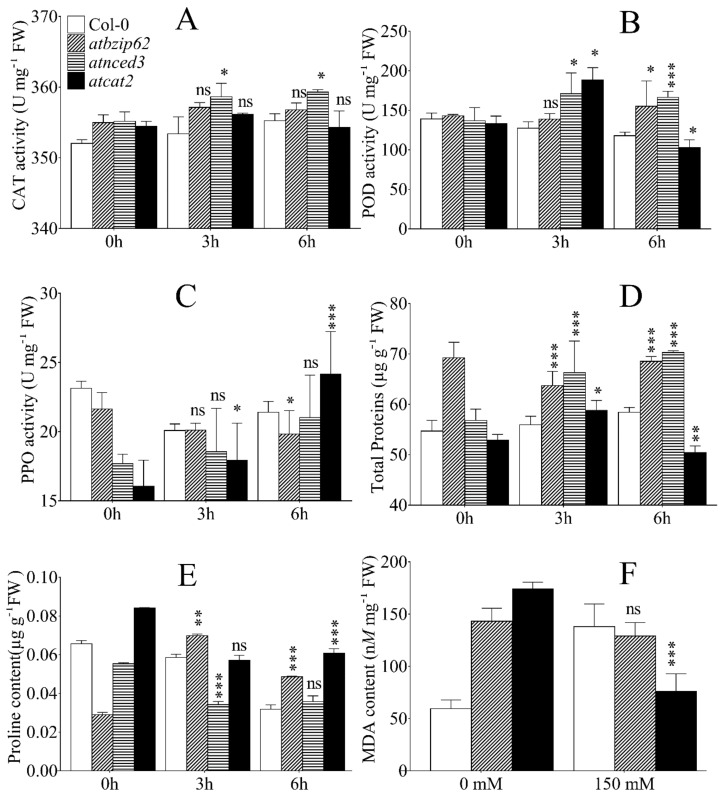
Antioxidant enzyme activity in response to 150 m*M* NaCl in *Arabidopsis* genotypes (Col-0 wild type, *atbzip62*, *atnced3*, and *atcat2* mutants) subjected to salt stress conditions at the rosette stage. (**A**) Catalase (CAT) activity. (**B**) Peroxidase (POD) activity. (**C**) Polyphenol oxidase (PPO) activity. (**D**) Change in total protein content. (**E**) Change in proline content. Data in A–E were collected 3 h and 6 h after stress. (**F**) Malondialdehyde (MDA) content as a measure of lipid peroxidation for Col-0 wild type, *atbzip62*, and *atnced3* exposed to no salt or to 150 m*M* NaCl. The data points are mean values of triplicate assays, while error bars indicate ±SD. * *p* < 0.05; ** *p* < 0.01; *** *p* < 0.001; ns, non-significant.

**Figure 6 ijms-21-01726-f006:**
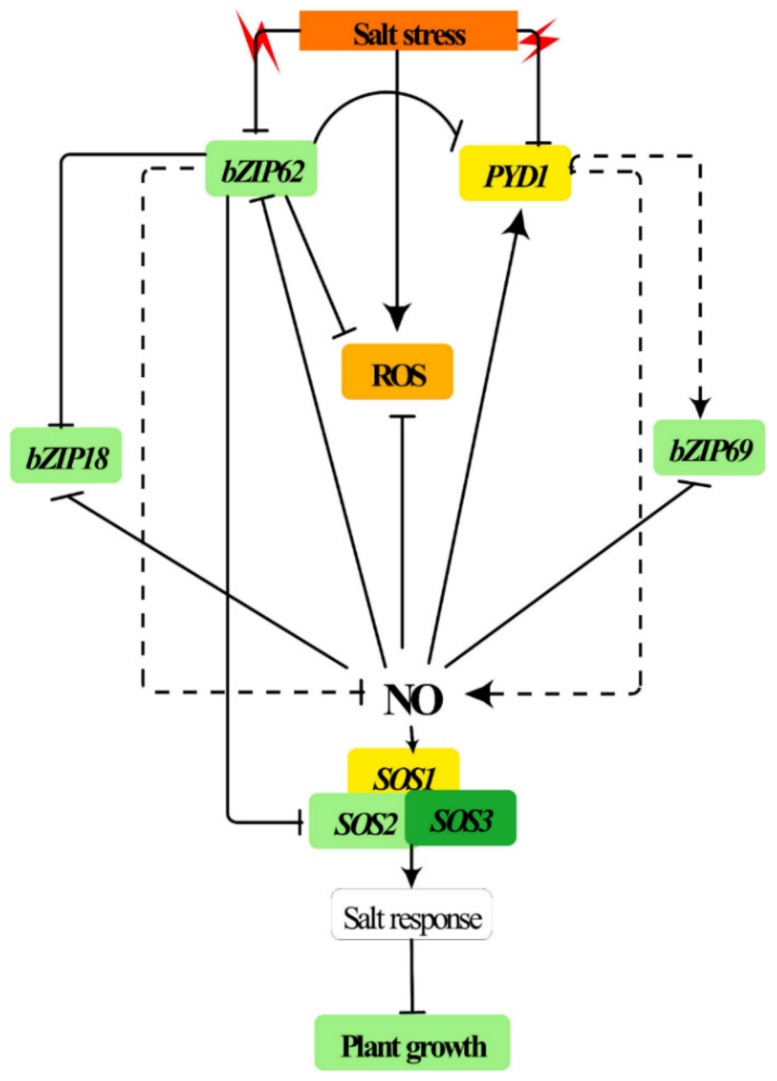
Simplified model of the transcriptional regulation of three salt overly sensitive (SOS) pathway genes in response to salinity in *Arabidopsis*. *AtbZIP62* is shown to control the expression of *SOS* genes in *Arabidopsis* plants exposed to salt stress. The loss-of-function mutant atbzip62 significantly upregulates *SOS* genes. *AtbZIP18* and *AtbZIP69* are genes coding bZIP transcription factors proposed to have an interplay with the *AtbZIP62* under abiotic stress conditions. Upon salt stress perception, plants activate various metabolic and physiological processes, which include the induction of signaling components that interact with each other to provide the proper response to stress tolerance. During this event, plants redirect their resources and allocate them to protect against the stress, affecting their growth and development. ROS, reactive oxygen species. NO, nitric oxide.

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
