# Peer review of "Salinity Stress-Mediated Suppression of Expression of Salt Overly Sensitive Signaling Pathway Genes Suggests Negative Regulation by AtbZIP62 Transcription Factor in Arabidopsis thaliana"

_ijms, 2020, doi:10.3390/ijms21051726_

Round 1

Reviewer 1 Report

The manuscript is worth publishing, however it requires some corrections.

The abstract does not explain the purpose of the work and differences between the examined mutants. This information is also missing in the Introduction. They are included only at the end of the manuscript in Materials and methods. It is necessary to clarify the term "loss-of function mutant". What functions are these? Abstract presents the results and conclusions enigmatically. Only generalities. If there was a change in catalase activity or proline accumulation, did they increase or decrease? What does it mean in the mutants' response to salinity?

Introduction

It should be clearly indicated which processes occur within the Salt Overly Sensitive pathway. In the Introduction, it is necessary to discuss each mutant studied, why these forms were chosen? There is no clearly defined purpose of the work.

Results

How was CDF determined? This information is also missing in the Methods. References are not quoted in the Results.

Conclusions

The presented chapter does not contain any conclusions. They should clearly outline to the reader the meaning of the results obtained from this work.

Author Response

Salinity stress-mediated suppression of expression of Salt Overly Sensitive signaling pathway genes suggests negative regulation by AtbZIP62 transcription factor in Arabidopsis thaliana

Nkulu Kabange Rolly1,2, Qari Muhammad Imran1, In-Jung Lee3 and Byung-Wook Yun1,*

Manuscript ID: ijms-727167

Point-by-point reply

Reviewer 1

1

The abstract does not explain the purpose of the work and differences between the examined mutants. This information is also missing in the Introduction. They are included only at the end of the manuscript in Materials and methods.

It is necessary to clarify the term "loss-of function mutant". What functions are these?

Abstract presents the results and conclusions enigmatically. Only generalities. If there was a change in catalase activity or proline accumulation, did they increase or decrease? What does it mean in the mutants' response to salinity?

We are thankful to the anonymous reviewer for giving time to our manuscripts. All the comments were genuine, and we have tried our best to address all of them. The new changes have been incorporated through track changes for visibility in the revised manuscript and highlighted green.

The abstract has been extensively revised, brief information of mutant line have been added to the revised MS. The word loss-of-function was replaced with knocked out mutant which refers to T-DNA insertion line

As mentioned above, the abstract has been revised as per suggestions by the worthy reviewer, we hope it will better than the previous version and would clearly present the intended meaning.

2

Introduction

It should be clearly indicated which processes occur within the Salt Overly Sensitive pathway. In the Introduction, it is necessary to discuss each mutant studied, why these forms were chosen? There is no clearly defined purpose of the work.

The introduction and brief mechanism of SOS pathway has been added to the revised MS now line 69-71, 74-79.

The detail of mutant line used have been given in the introduction section in the revised MS line 95-97. Also the objective of the study is briefly mentioned in the introduction section last paragraph

3

Results

How was CDF determined? This information is also missing in the Methods. References are not quoted in the Results.

The CDF was calculated as the number of developed green seedlings relative to sown/germinated initially but didn’t grow as described earlier by [1, 2]. We have added this to the results section line 110-112, and materials and methods section line 416-418 in the revised MS.

4

Conclusions

The presented chapter does not contain any conclusions. They should clearly outline to the reader the meaning of the results obtained from this work.

The conclusion section has been rephrased to meet the worthy reviewer’s recommendations (L.463-476)

  1. Imran, Q. M.; Hussain, A.; Lee, S.-U.; Mun, B.-G.; Falak, N.; Loake, G. J.; Yun, B.-W., Transcriptome profile of NO-induced Arabidopsis transcription factor genes suggests their putative regulatory role in multiple biological processes. Scientific Reports 2018, 8, (1), 771.
  2. Yun, B.-W.; Feechan, A.; Yin, M.; Saidi, N. B. B.; Le Bihan, T.; Yu, M.; Moore, J. W.; Kang, J.-G.; Kwon, E.; Spoel, S. H.; Pallas, J. A.; Loake, G. J., S-nitrosylation of NADPH oxidase regulates cell death in plant immunity. Nature 2011, 264-268.

Reviewer 2 Report

Introduction.

L.68. Add the references about transgenic plants overexpressing SOS genes.

Matherials and Methods.

How many seeds/plants were used to test germination and salt stress tolerance?

Subsection 4.8. What wavelengths were used to calculate pheophytin contents? The OD values are presented only for chlorophyll and carotenoids.

The drought-treated plants were mentioned in the manuscript (L.124, 373), but this experiment was not described.

L.374. The atpyd1-2 genotype is mentioned, but it is not described in the subsection 4.1.

At what time point of the experiment the samples for RNA isolation, proline and enzyme activity assays were selected?

The experiments of exposure to NaCl at rosette stage during 0, 3, 6 h (section 2.9) and 10 days (section 2.7) are not described.

Results.
Subsection 2.5. Why did you measure the transcriptional levels of glutamate synthase genes, rather than glutamine synthetase genes (the GS/GOGAT cycle)?

Salt stress can affect the water content in plants. Why the protein and proline contents, enzyme activity were calculated on the FW basis not DW?

In Fig. 2G heading should be AtSOS3.

What is the difference between Pheophytins a + b (Fig. 4G) and Total pheophytins (Fig. 4H)? The figures are very similar, but the concentration of Total pheophytins is approximately 4 times lower.

The proline concentration was measured in μg/g FW (L.384-385), but the data are presented in mg/ml (Fig. 5E).

Fig. S1. There are no statistical analyses.

Fig. S2. Delete the missing atgsnor 1-3 genotype from figure caption.

No uniformity in captions of Fig. S1, S2 (atbzip62, …) and Fig. S3, S4 (bzip62, …).

Number of independent replicates (n) it is necessary to indicate in Figures 1, 2, 4, 5, S1, S3, S4.

Table S1. No primers for AtGLS1 and AtGLS2 genes (Fig. 2) in the table.

References.

The species names should be italicized.

Author Response

Reviewer 2

1

Introduction.

L.68. Add the references about transgenic plants overexpressing SOS genes.

We are thankful to the reviewer for valuable suggestion that will surely improve the manuscript.

The relevant reference has been added to the revised manuscript now line 75 as [1]

2

Materials and Methods.

How many seeds/plants were used to test germination and salt stress tolerance?

Subsection 4.8. What wavelengths were used to calculate pheophytin contents? The OD values are presented only for chlorophyll and carotenoids.

The drought-treated plants were mentioned in the manuscript (L.124, 373), but this experiment was not described.

L.374. The atpyd1-2 genotype is mentioned, but it is not described in the subsection 4.1.

At what time point of the experiment the samples for RNA isolation, proline and enzyme activity assays were selected?

The experiments of exposure to NaCl at rosette stage during 0, 3, 6 h (section 2.9) and 10 days (section 2.7) are not described.

We are sorry for the inconvenience caused, The following text was included in the revised MS now L.354 and 371:” For each genotype, 30 seeds were sown, 10 per replication”.

L. 451-452, the following OD values for pheophytin were included:” 665, 653 and 470 nm for pheophytin a, b, and total pheophytin”.

We sincerely apologize for the wrong typing, We have now corrected this in the revised manuscript now L. 24. We intended to say NaCl-treated plants of Col-0 instead of drought treated plants of Col-0.

Authors would like to apologize for the inconvenience. atpyd1-2 genotype was used in an another study and was mistakenly typed here Therefore, we deleted it from the revised MS now.

We have now mentioned the time points for samples collected for gene expression analysis, proline and enzyme activity assays in the revised manuscript now L.373  (0, 3 and 6h).

Authors included the description related to NaCl treatment at rosette stage in the section 4.2 (L.371-376.

3

Results.
Subsection 2.5. Why did you measure the transcriptional levels of glutamate synthase genes, rather than glutamine synthetase genes (the GS/GOGAT cycle)?

Salt stress can affect the water content in plants. Why the protein and proline contents, enzyme activity were calculated on the FW basis not DW?

In Fig. 2G heading should be AtSOS3.

What is the difference between Pheophytin a + b (Fig. 4G) and Total pheophytin (Fig. 4H)? The figures are very similar, but the concentration of Total pheophytins is approximately 4 times lower.

The proline concentration was measured in μg/g FW (L.384-385), but the data are presented in mg/ml (Fig. 5E).

Fig. S1. There are no statistical analyses.

Fig. S2. Delete the missing atgsnor 1-3 genotype from figure caption.

No uniformity in captions of Fig. S1, S2 (atbzip62, …) and Fig. S3, S4 (bzip62, …).

Number of independent replicates (n) it is necessary to indicate in Figures 1, 2, 4, 5, S1, S3, S4.

Table S1. No primers for AtGLS1 and AtGLS2 genes (Fig. 2) in the table.

Authors appreciate the concern expressed by worthy reviewer about GS/GOGAT cycle. Authors selected glutamate synthase, which is also referred to as GOGAT, for its role in the nitrate assimilation as one of the NH4 carriers. Under salinity stress, nitrate assimilation is expected to be affected due to water unavailability.

We appreciate the concern expressed by worthy reviewer and bringing it to our notice. Actually the method for proline contents that was possible in our laboratory was the colorimetric method which measures proline based on FW basis. However, we shall consider the point raised by worthy reviewer to optimize protocol for the dry weight basis as well for our future studies.

We appreciate the meticulous screening of the manuscript by worth reviewer. The suggests changes have been included in Figure 2G in the revised manuscript.

Figure 4G is regarded as the content of both pheophytin a and b in the plant extract. However, Figure 4H is the measurement Total Carotenoids determined by deduction of the relative absorption of pheophytin a and b from the absorbance read at 470 nm followed by division by the absorption coefficient of total carotenoids at 470 nm.

Here, the total sum of carotenoids relative to the pheophytin a and b was determined in a pigment extract together with pheophytin a and b, provided that chlorophylls undergo pheophytinization in the presence of acid environment. Reports indicate that small amounts of pheophytin a have always been found in plant pigment extracts and seem already to be present in the intact leaf. In fact, pheophytin a is a functional component of the reaction center of photosystem II, where it plays a role in the charge separation and electron transfer.

Pheophhytin a and b are formed under the influence of the endogenous organic acids during pigment extract preparation. This is particularly needed in the case of plants which are especially known to store organic acids in their vacuoles[2].

Authors apologize for the mistyping. Figure 5E label was corrected after checking the data.

The error bars was applied to the Figure S1 as suggested by worthy reviewer.

We are sorry for the inconvenience caused on our part and appreciate the deep screening of the manuscript by worthy reviewer to improve its technical content. atgsnor1-3 was deleted from the caption of Figure S2 as recommended.

We have now uniformed the captions of supplementary figures according to worthy reviewer’s suggestions

The statement: “data are means of triplicate” was included in the caption of all suggested figures.

The primers for AtGLS1 and 2 are included in the Table S1.

4

References.

The species names should be italicized.

All species names were italicized in the list of references as suggested by the worthy reviewer.

  1. Ma, D.-M.; Xu, W.-R.; Li, H.-W.; Jin, F.-X.; Guo, L.-N.; Wang, J.; Dai, H.-J.; Xu, X., Co-expression of the Arabidopsis SOS genes enhances salt tolerance in transgenic tall fescue (Festuca arundinacea Schreb.). Protoplasma 2014, 251, (1), 219-231.
  2. Harmut, A., Chlorophylls and carotenoids: pigments of photosynthetic membranes. Methods Enzymol. 1987, 148, 350-383.

Reviewer 3 Report

This manuscript described a comprehensive investigation on salt sensitive mutants in arabidospsis. Authors described transcriptional, physiological and biochemical differences between these mutants with their relative controls. Conclusions are convincing. However, the language and structure of manuscript need to be substantially improved. I have some suggestions for authors.

  1. In figure 1, the inhibition is only significant in 100mM, but in the manuscript you claimed 150mM as the most effective concentration and used for downstream analysis. Authors need a better description in here. Also, figure 1 is confusing, if you compared data within each group to control, why not labeled significant level in concentration 0mM?
  2. Result 2.3, “…investigated the transcriptional level” by which technology? qPCR should be specified here but not after. You listed a number of genes being tested here, but what did you test them and how did you pick these? Are those results reasonable? What are your interpretations for these results?
  3. I don’t quite understand why don’t you merge results 2.2 and 2.6 together?
  4. Result 2.7 is much better to merge with 2.2 and 2.6 or present it at the beginning to help readers understand the phenotype your observed.
  5. Figure 5, there is no need to give a different color label in figure 5F, please use consistent color labeling for all subpanels in figure 5.

Author Response

Salinity stress-mediated suppression of expression of Salt Overly Sensitive signaling pathway genes suggests negative regulation by AtbZIP62 transcription factor in Arabidopsis thaliana

Nkulu Kabange Rolly1,2, Qari Muhammad Imran1, In-Jung Lee3 and Byung-Wook Yun1,*

Manuscript ID: ijms-727167

Point-by-point reply

Reviewer 3

1

In figure 1, the inhibition is only significant in 100mM, but in the manuscript you claimed 150 mM as the most effective concentration and used for downstream analysis. Authors need a better description in here.

Also, figure 1 is confusing, if you compared data within each group to control, why not labeled significant level in concentration 0mM?

We are thankful to the anonymous reviewer for giving valuable time and appreciate the concern raised by worthy reviewer. We have tried our best to improve the manuscript in the light of reviewer suggestions. The changes suggested are highlighted pink.

150 mM was selected considering all screening data, not only CDF, such as germination pattern (Figure S1), shoot (Figure S3) and root (Figure S2, Figure S4) lengths.

We have compared the response to each genotype under salt stress compared to wild type to see a genotype-specific response. As under control conditions, there was no significant response among different genotypes therefore no significant label was put there.

2

Result 2.3, “…investigated the transcriptional level” by which technology? qPCR should be specified here but not after. You listed a number of genes being tested here, but what did you test them and how did you pick these? Are those results reasonable? What are your interpretations for these results?

L.133, by qPCR was specified as the technology use to measure the gene expression.

Authors appreciate the concern raised by worthy reviewer. The set genes used in the study were selected based on the prediction made in our previous studies. AtPYD1 is the ortholog of OsDHODH1 encoding an important protein catalyzing the 4th step of the pyrimidine biosynthesis pathway. We previously reported the inhibition of OsDHODH1 by nitric oxide donor (Sodium nitroprusside) [1]. We were interested to study the transcriptional regulation of this gene using Arabidopsis mutant line. As for AtbZIP18 and AtbZIP69, their candidate genes encoding transcription factors belonging to bZIP family included to study their transcriptional interplay with the AtbZIP62

3

I don’t quite understand why don’t you merge results 2.2 and 2.6 together?

The results have been merged as suggested by the worthy reviewer.

4

Result 2.7 is much better to merge with 2.2 and 2.6 or present it at the beginning to help readers understand the phenotype your observed.

Results 2.2 and 2.6 were merged together while 2.7 was brought to 2.3 as suggested by the worthy reviewer.

5

Figure 5, there is no need to give a different color label in figure 5F, please use consistent color labeling for all subpanels in figure 5.

Done as suggested by the worthy reviewer

  1. Rolly, N. K.; Lee, S.-U.; Imran, Q. M.; Hussain, A.; Mun, B.-G.; Kim, K.-M.; Yun, B.-W., Nitrosative stress-mediated inhibition of OsDHODH1 gene expression suggests roots growth reduction in rice (Oryza sativa L.). 3 Biotech 2019, 9, (7), 273.

Round 2

Reviewer 2 Report

The manuscript has been improved and can be accepted for publication in this form.